# Does Distance Affect the Role of Nonlocal Subsidiaries on Cluster Firms' Innovation? An Empirical Investigation on Chinese Biotechnology Cluster Firms

**Xiyao Xiang [1],\* and Wei-Chiao Huang [2]**

[1] Economics and Management School, Xi'an University of Technology, Xi'an 710048, China
[2] Economics Department, Western Michigan University, Kalamazoo, MI 49008, USA;
   wei-chiao.huang@wmich.edu
\* Correspondence: xxy928215@xaut.edu.cn; Tel.: +86-029-6266-0219

**Abstract:** As translocation strategy has been pursued by cluster firms, two types of nonlocal subsidiaries, nonlocal manufacturing subsidiaries (NMS) and nonlocal R&D subsidiaries (NRS) contribute to their holding firms' innovation in different ways. Prior studies have not paid much attention to the role of NMS and NRS, and how their effects are contingent on distance. To address this gap, this paper assesses the contribution of NMS and NRS on cluster firms' innovation performance respectively and ascertains the moderating effect of geographical distance and social distance. The empirical investigation is conducted using a sample of 79 Chinese cluster firms. Our results indicate that both NMS and NRS have positively influenced cluster firms' innovative performance. Moreover, geographical distance negatively moderates both the role of NMS and NRS. On the other hand, social distance only increases the influence of NRS on their holding firms' innovation. Thus, spatial distance may hamper knowledge acquisition through NMS and NRS while loosely connected NRS would contribute more to their parent companies in local clusters. Our research contributes to the literature on cluster firms' relocation strategy by clarifying the distinct role of NMS and NRS and recognizing the contingent effect caused by geographical and social distance.

**Keywords:** industrial clusters; nonlocal subsidiaries; innovation performance; relocation; geographical distance

## 1. Introduction

Over the past decades, much attention has been devoted to industry territorial agglomeration [1] and its contribution to the economic development of emerging countries [2]. One of the central tenets of present literature on industrial clusters is that geographical proximity facilitates knowledge flows and learning processes [3,4], which are crucial to the success of clusters and firms' innovation [5,6]. However, some researchers also point out that clusters may suffer from technological lock-in due to over-embedding in local network [7–9] or paradox of proximity [10,11], i.e., too much closeness generates obstacles to intra- and extra-cluster knowledge exchange [12]. To avoid the shortcoming of over-closeness, scholars suggested that cluster firms should reorganize their present knowledge networks via introducing nonlocal knowledge ties [6]. Most of the work in this area view relocation as a solution for firms facing cost pressure and innovative dilemma [13] or as a natural evolution process for clusters [14,15], especially for Chinese cluster firms, which have long been labeled as a proficient imitator. This is exemplified by the cross-border mobilization of Taiwan IT firms to mainland China since the 1990s [16] and the recent movement of manufacturing firms from Pearl River Delta to

inland provinces in China [17]. Some recent studies found that cluster firms tend to become active in investing on nonlocal manufacturing subsidiaries (NMS) or nonlocal R&D subsidiaries (NRS) during the expanding stage of clusters [18]. Through NMS, cluster firms can absorb novel manufacture knowledge by co-working with manufacturers or clients in host regions. Besides that, NRS are also key platforms that hold crucial nonlocal knowledge linkages, which play an irreplaceable role in cluster firms' innovation [19]. However, these two types of subsidiaries contribute to cluster firms' innovation with distinct types of knowledge and pathways. Through NMS, cluster firms are more likely to gain access to practical knowledge in work sites and that may stimulate imitation or incremental innovation in new technology directions. In contrast, NRS, which are designed as R&D collaborative platforms, may push cluster firms to focus more on developing new technology or creating novel knowledge. In the present literature, not much attention has been paid to distinguish the distinct role of nonlocal subsidiaries on their holding companies' innovation.

Furthermore, NMS and NRS normally focus on different types of knowledge in terms of the complexity and novelty and as such would develop diverse knowledge absorbing and sharing systems. Very few studies have investigated how geographical and social distances, which are crucial in both intra- and inter-regional knowledge diffusion [20], affect a nonlocal subsidiaries' role in cluster innovation [18,21].

In an attempt to bridge these gaps in the literature, this paper, drawing from knowledge diffusion theory and synthesizing perspectives of economic geography and social network, develops a framework to answer two questions: First, whether both of NMS and NRS enhance cluster firms' innovation performance? Second, how geographical and social distances influence different subsidiaries' role. Our research contributes to present literature in two ways. First, we go beyond prior studies that solely focused on one types of nonlocal subsidiaries and confirm the role of both NMS and NRS on cluster firms' innovation. Second, our empirical results clarify the moderate effect of geographical distance and social distance on the roles of different types of subsidiaries, which extends the present literature on extra-cluster knowledge searching and nonlocal collaboration. We use a 79-firms sample from China where industrial clusters have been developed for decades [22]. Currently, many clusters in eastern provinces and other advanced regions of China are experiencing pressures to upgrade from labor-intensive manufacturers to innovative products providers, and thus have started to relocate part of their capacities to less advanced provinces of China [17]. This ongoing development would provide us with enriched data to conduct empirical study on nonlocal subsidiaries related topics. The rest of the paper is organized as follows. Section 2 summarizes related theory and prior studies from which we propose four hypotheses and develop a framework to explore the influence of NMS and NRS on cluster firms' innovative performance. Moreover, the moderating role of geographical distance and social distance are also discussed. Section 3 presents our data collection method and describes the 79-firms sample and variables. Section 4 presents empirical analysis and discusses the estimation results. The concluding section summarizes the study and provides policy implications.

## 2. Theory and Hypotheses

### 2.1. Nonlocal Subsidiaries and Innovation

Setting up nonlocal branches represents the effort taken by cluster firms to counter rising labor costs [23] and to avoid cluster myopia [24]. Through relocating manufacturing capacity or R&D department fully or partially, cluster firms become closer to new external knowledge sources. However, NMS and NRS contribute to different aspects of their holding firms' innovative activity.

From a knowledge diffusion perspective, the enhancement effect of NMS on their holding firms' innovative capability is realized in two ways. First, NMS increase the opportunity of imitation through "learning by watching" [25], which stimulates incremental innovation. Cluster firms' NMS have more chance and also take lower costs as well as less risk to share product designs through observing the creation process of new technology on spot or acquiring new knowledge from skilled employees in the

host region. In addition, NMS can benefit from knowledge spillovers or "mimetic isomorphism" [26], and develop novel technological competence by embedding in a local business network, interacting with nonlocal consumers, suppliers, rivals as well as public innovation research institutions [20,27] and that also expands cluster firms' knowledge searching breadth [28]. By translocating manufactural departments, cluster firms not merely benefit from the cost advantage but also accessing to new technical resources [17] from different regions and industries [29]. Though most of NMS are more concerned with manufacturing skills or know-how, and have less chance to access core knowledge and less capable of absorbing advanced technology, still cluster firms could benefit from NMS's imitation activities, which have been found in some cases to perform better than original inventors [30]. Therefore, NMS can provide cluster firms with new skills, which may help them generate new ideas in designing and manufacturing new products.

Second, NMS facilitate transfer of some sorts of sticky knowledge such as the way to improve product quality and efficiency of manufacturing processes [31]. The stickiness nature of knowledge, typically characterized as highly tacit and context-dependent, necessitates firms to collaborate with distant problem solvers directly to share working experience and know-how [32]. By working together, the inflows of such highly site-dependent knowledge would enhance firms' capacity in tackling technological difficulties and add in the innovation process with the contribution of valuable new ideas from distant problem solvers [33]. Thus, NMS can provide cluster firms with novel practice knowledge, which in some case is more valuable than basic scientific findings [34].

In summary, with the assistance of NMS, cluster firms might achieve better innovative performance via stimulating imitation and obtaining problem-solving knowledge. Thus, we posit that:

**Hypothesis (H1).** *cluster firms that have nonlocal manufacturing subsidiaries (NMS) achieve higher innovation performance.*

Through relocation, cluster firms also initiate NRS, such as common innovative centers, R&D institutes or laboratories, to bond with nonlocal partners for knowledge creation. Unlike the role of NMS, NRS are more important in building cluster firms' novel knowledge system. According to existing studies in literature on knowledge diffusion and creation, NRS' contribution is twofold. First, cross-boundary innovation conducted by NRS assists cluster firms in renewing their existing knowledge system [35] and innovation capacity [36,37]. Through co-innovation with partners beyond clusters, heterogeneous knowledge could be created by and transferred from NRS. In addition, since NRS usually innovates on proprietary knowledge basis and in highly specific technology domain, the contributions of NRS to holding firms are to a large extent associated with the variety of knowledge trajectory. Thus, cluster firms would be more innovative [38] by integrating distinct technological streams from various NRS. In addition, cluster firms commonly prone to set up NRS jointly with extern organizations. There are often various types of NRS contribute to their innovative activity. Therefore, the more and larger a firms' NRS group, the greater improvements in innovation the firm will achieve.

Second, NRS has advantage in stimulating radical innovation. Literature in temporary proximity argued that knowledge creation of cluster firms could come from short term contacts [39,40] such as conferences, workshops and exhibitions [41] due to the exposure of cluster firms to external knowledge sources. Following this logic, cluster firms will benefit more through their NRS, which is a sort of regular or semi-permanent organization and has obviously greater advantage in absorbing systematic knowledge via recurring and regular knowledge exchange. Such substantial contacts and interactions are more helpful for cluster firms to integrate and recombine various external knowledge elements. Therefore, radical innovation is more likely to occur during co-invention process conducted by NRS and nonlocal organizations.

With all these arguments considered, we propose the following hypothesis:

**Hypothesis (H2).** *cluster firms with more nonlocal R&D subsidiaries (NRS) can attain higher innovation performance.*

### 2.2. Geographical and Social Distance and the Role of Nonlocal Subsidiaries

As discussed above, the reason that NMS and NRS can contribute to their holding firms' innovation is largely attributed to their distinct learning pattern. It does not imply that cluster firms would benefit equally from subsidiaries in different locations. NMS tend to undertake down-stream technology solving or manufacture process adjustment, which merely stimulates incremental technology improvement rather than original invention. Distance, in both spatial and social aspects, impedes knowledge diffusion of both NMS and NRS, as contended by some studies.

Based on the theory of economic geography, scholars have provided abundant evidence about the influence of geographical distance on knowledge exchange and innovation [42–45]. It is commonly accepted in this stream of literature that the effect of geographical distance cannot be easily smoothed [46] or remedied by other dimensions of distance (cognitive, social as well as cultural) or even temporary spatial proximity with the aid of advanced communication technology [39] due to a variety of factors such as the concentrated skilled labor pool [47,48], frequent knowledge exchange [42,49] as well as dense local social relationships [50,51], which are still essential and un-substitutable elements for innovation activities.

In this vein, closer geographical distance between NMS/NRS and their parent company would enhance the efficiency and accuracy of knowledge transfer from "learning by watching" and co-invention with partners beyond clusters. For one thing, the outcome of knowledge exchange heavily relies on the willingness of knowledge holders to share their experience and know-how [52]. Geographical distance plays a significant role in affecting the mobility of skilled labors and managers and so the building of greater mutual trusts [53,54] as well as reciprocity [55]. Thus, with the advantage of spatial nearness, NMS/NRS could get a higher level of support from the top management of their holding firms who encourages them to transmit heterogeneous manufactural experience constantly and provide feedback to parent company's inquiry expeditiously.

Further, external skills acquired at the workplaces can not automatically yield innovation. Giuliani (2005) pointed out that the capacity to absorb external knowledge and then integrate it into the firm's inner knowledge base is crucial to the growth of clusters' knowledge repository [56]. Besides that, situated learning undertaken by NMS/NRS specially requires physical co-presence and the learning outcomes are constrained by geographical distance. Through face-to-fact contacts and communication, highly context-dependent knowledge can be codified and shared among actors from distinct technical background. This effect is particularly evident in the context of NRS. Hence, diversely distributed research departments may not be an optimal arrangement for cluster firms to absorb tacit knowledge. To collect knowledge from an unfamiliar environment, similar technological background [57] and organizational institutions or contexts [58] are considered as crucial elements of absorptive capacity. However, geographical distance may pose barriers to smooth formation of the firm-subsidiary knowledge linkages and diffusion of tacit knowledge [24]. As spatial distance grows, technology trajectory becomes more diverse and discrepant, forcing cluster firms to put more energy onto adjusting their production system to cope with idiosyncratic technological systems. In that case, NMS/NRS located near their parent firm are more capable of knowledge absorption from user-producer interaction so as to generate higher innovative performance [59,60].

In sum, NMS/NRS in closer proximity are more likely to transfer knowledge precisely and systematically back to their holding firms and so will contribute more to the innovation performance of parent company. Based on the above arguments, we postulate that the effect of NMS/NRS on cluster firms' innovative performance is contingent on geographical distance.

**Hypothesis (H3$_A$).** *geographical distance negatively affects the relationship between NMS and innovation performance.*

**Hypothesis (H3$_B$).** *geographical distance negatively moderates the relationship between the number of NRS and innovation performance.*

Contrary to traditional theory of agglomeration economy, the literature on multiple proximity argues that geographical closeness can be an obstacle to innovation over time [12,61] and plays a diminishing role [62,63] due to the proximity paradox [10,11] and advanced communication technology [64]. Following this stream of research, firms could acquire more novel knowledge from remote organizations and thus social distance, an invisible range between actors in terms of social relationships, can reduce the geographical distance between partners [65] and that may actually explain better the micro mechanism of knowledge flows [66,67]. Moreover, actors with short social distance have an obvious advantage in running long-term co-invention projects since social relationships will sustain even when previously co-located individuals are separated, i.e., "gone but not forgotten" [68].

For NMS, accommodating to a distinct social context and developing trustworthy bridge-makers [69] are crucial for them to learn by watching. Embedding into a nonlocal market can facilitate NMS to resolve uncertainty [70] but also inevitably engender social distance between NMS and their holding firms in remote clusters. Such social dissimilarity impedes knowledge diffusion from NMS to parent companies. Researches from a social capital perspective have noted that frequent interpersonal contacts and prior collaboration ties or bridging ties [29] largely determine the level of trust [29,71], which is heavily associated with the extent of actors involving in co-invented relationship and the efficiency of learning and knowledge diffusion [72]. Formal and informal extra-cluster relationships, in both the organizational and private level, are helpful in shaping close collaboration networks [73], which facilitate cross-cluster knowledge diffusion. Specifically, continuous contacts are conductive to removing obstacles in diffusing and sharing know-how and operational experience as well as various types of context-dependent knowledge. Thus, intimate and cohesive social relationships among experienced workers and technicians would enhance the possibility of successful knowledge transfer from NMS to their holding firms. Following this logic, this study conjectures that social distance will diminish the positive impact of NMS on their holding firms' innovation.

In the context of NRS, knowledge acquired is creative in nature and mostly relies on frequent contacts during co-invention with partners in the hosting region rather than merely "watching". Unlike the knowledge acquiring process that NMS conducts, NRS have to tackle with more complicated and intangible knowledge, which cannot be secured by gathering in similar places. Knowledge as a relational source [74] is inevitably related to social context in specific places and has to be consistently invested in the social relationship so as to cultivate relational proximity [75] and cultural proximity [76]. To assimilate into a local community in a remote region where language, norms, cultures as well as visions are rather different, NRS have to spend more energy to fit its style of innovation to new circumstance and so could become socially dissimilar to their holding firms, which are heavily embedded in local social context. The literature on social proximity finds that being socially close is more important in building collaboration ties compare with geographical distance [77], since intense collaboration relationships with long-term duration can significantly reduce the uncertainty of co-invention, improve the efficiency of knowledge sharing [78,79] as well as enhance innovation performance [80]. According to this logic, larger social distance may generate obstacles in knowledge diffusion and will hamper co-invention activities between cluster firms and their NRS. Consequently, it is reasonable to expect that the role of NRS will be sensitive to the closeness of co-invented relationships.

In summary, we propose the following hypotheses:

**Hypothesis (H4$_A$).** *social distance negatively affects the relationship between NMS and innovation performance.*

**Hypothesis (H4$_B$).** *social distance negatively affects the relationship between NRS and innovation performance.*

The conceptual framework discussed in the above and the resulting hypotheses are shown in Figure 1.

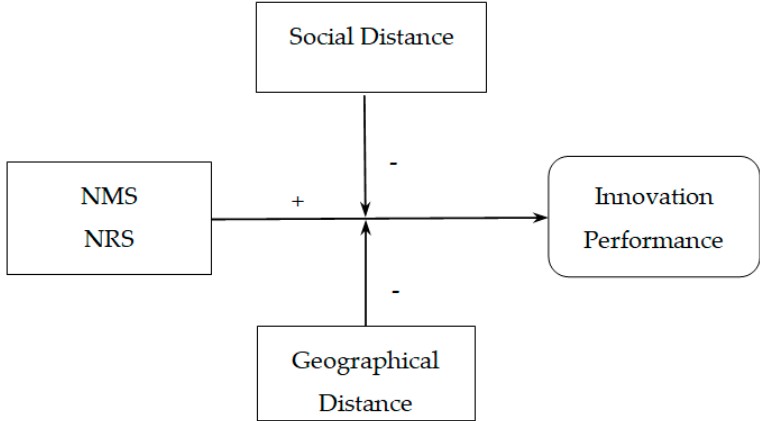

**Figure 1.** Theoretical model.

## 3. Data and Methodology

### 3.1. Sample

We collected data by interviewing, with a semi-structured questionnaire, general managers or senior managers who are in charge of firms' R&D related issues in two biotechnology clusters, including over 400 firms, in Nanjing and Xi'an metropolitan areas in China. More specifically, these firms are mainly located in Jiang Su Life Park (Nanjing) and Xi'an Hi-Tech industries Development Zone where management committees are in charge of clusters' daily management and public service. We chose our research sample in the biotechnology industry for two reasons. First, the biotech industry is highly motivated to engage in innovation as this is an industry where knowledge becomes obsolete quickly [81]. Since there are broad subsectors in this industry, firms in specific segments have various collaboration choices and location strategies [82]. Thus, firms in biotech clusters tend to be active innovators with strong incentive to learn from local and nonlocal partners. As such, extra-cluster innovative cooperation and nonlocal subsidiaries are also expected to be prevalent in biotech industry. Second, as a new booming industry, biotech clusters receive strong support from local governments in both Nanjing and Xi'an where more than 400 firms are congregated in two cities, which provide reasonable sample size for our research.

To ensure all the questionnaire items are totally understood, we conducted an on-site survey rather than simply sending e-mails. Our sample was randomly chosen from two clusters and the name and basic information of firms were acquired from the local High-tech Zone Management Committees in Nanjing and Xi'an. Our on-site survey was assisted by officers from local management committees and that helped us to get complete data from our interviewees. Moreover, to avoid missing values and an invalid questionnaire, our data was also checked carefully during the survey. In all, 132 managers were interviewed and provided us with R&D and collaboration related information. Since our main focus was the effect of nonlocal relationships on innovation, those firms that did not have nonlocal innovative partners in the recent five years were excluded. Our final sample consisted of 79 cluster firms from both the two regions, which meant that 59.84% of interviewees provided usable responses. Table 1 provides a basic information summary of the sample.

In this survey, we defined key partner as the one with which the firm has had collaboration experiences for at least one year and have generated new products or patents together in the past five years.

**Table 1.** Basic description of 79 sample firms.

|  | **Proportions** |
|---|---|
| **Ownership of firms** |  |
| State-owned | 5.3% |
| Private | 94.7% |
| **Asset size** |  |
| Over 10 million yuan (RMB) | 12.5% |
| 2–10 million yuan (RMB) | 41.67% |
| Below 2 million yuan (RMB) | 45.83% |
| **Stage of Development** |  |
| Start-ups | 14.6% |
| Matured firms | 85.4% |
| **Geographical distribution of key partners** |  |
| Local | 4% |
| Domestic Nonlocal | 83% |
| Foreign Nonlocal | 13% |

*3.2. Measurement of Variables*

3.2.1. Dependent Variable

Innovation performance. In the literature, there are a few variables that have been proved to be effective indicators of innovation performance. Among them, the amount of patents granted to a firm has often been adopted to gauge firms' innovation performance [83,84]. However, only one third of firms in our sample announced that they have patents authorized by the State Intellectual Property Office of China or similar offices in foreign countries. Besides that, the amount of patents granted to firms in our sample exhibits an extremely unbalanced distribution, which may induce bias in estimation. Hence, patent number is not a proper indicator of innovative performance in this case. We synthesized measurements used in prior studies [85,86] to come up with a composite index of innovative performance formed by four principal components, i.e., 'profits earned from new products achieved our goals', 'market share earned from new products met our expectation', 'we were able to proceed with new products R&D according to our schedule' and 'the complaint rate of our new products does not exceed our tolerance level'. Each component was measured using responses graded on a five-point Likert scale anchored from (1) 'strongly disagree' to (5) 'strongly agree'. Thus, the higher the value registered, the greater the innovative performance. To ensure accuracy of this measurement, we discussed with a group of managers in pretest to make sure they could totally understand all the questions and revised the expression of some items according to interviewees' suggestions. Then, an on-site survey, associated by officers from local High tech Zone Management Committee who had intimate contacts with sample firms, was conducted to make sure our data were in line with the real situation of firms. Based on the data collected, we calculated the Cronbach's α and conducted factor analysis. The results are summarized in Table 2.

**Table 2.** Cronbach's α and factor loadings.

| Items | Factor Loadings | Cronbach's Alpha |
|---|---|---|
| **InnovationPerformance** |  | 0.841 |
| Profits earned from new products achieved our goals | 0.766 ** |  |
| Market share earned from new products met our expectation | 0.786 ** |  |
| We were able to proceed with new products R&D according to our schedule | 0.768 *** |  |
| The complaint rate of our new products does not exceed our tolerance level | 0.762 ** |  |

*** $p < 0.01$, ** $p < 0.05$, * $p < 0.1$.

Table 2 shows that Cronbach's $\alpha$ for innovative performance was greater than the critical value of 0.7, reflecting a high level of internal consistency. The KOM (Kaiser–Meyer–Olkin measure) was 0.841, which indicates that it is suitable to conduct a factor analysis. The factor loadings from factor analysis for all four components are shown in Table 2. They were all statistically significant and larger than 0.7, indicating convincingly that we could measure our dependent variable based on these four items.

### 3.2.2. Explanatory Variables

Nonlocal manufacturing subsidiaries (NMS). Through NMS, cluster firms develop links to new knowledge sources. According to prior researches [18,21], holding ties constructed via translocation or nonlocal searching could benefit cluster firms' innovation. Our survey reveals that setting up production segments in other region required large financial support from the holding firms, and most of the firms we interviewed have only one extra-cluster plant. Thus, to detect the role of NMS on innovation performance of cluster firms, we used a dummy variable approach to measure NMS. Specifically, we asked the managers whether or not they had established new production lines, plants or factories, solo-invested or co-invested, beyond the region of the cluster. We defined nonlocal as places out of the range of the city where the firm was located. Joint ventures that did not take part in production were not included. This variable was coded 1 if a cluster firm had qualifying production sectors in different regions, and 0 otherwise.

Nonlocal R&D subsidiaries (NRS). The biotechnology industry is a typical high technology sector with intensive external collaborations, which involve various kinds of R&D subsidiaries, such as shared R&D institutes, laboratories as well as joint research centers. Since distinct types of NRS generate a variety of influence to their holding firms, and firm with more NRS may benefit more from nonlocal knowledge sources, it is inappropriate to use a dummy variable to reflect the degree of its involvement in trans-locational co-invention. Considering that most of our sample firms had only one sort of NRS, i.e., joint research center, and the focus of this research was on distinguishing NRS' role, we measured NRS by counting the number of R&D subsidiaries involved in external collaborations.

Geographic distance. This variable is to reflect the distance between cities that a cluster firm is located and the locus of its nonlocal subsidiaries. Prior studies have measured this variable by physical distance [87,88] or territory border [89]. However, that approach may not yield accurate measurements due to the different traffic condition among distinct regions in China. In general, the infrastructure condition is better in eastern China, which is more advanced in economic development than the West. Currently, high-speed railways and inter-city subways, labeled as faster and cheaper public traffic, become the most convenient and frequently used transport for inter-city travels in China. Nevertheless, there are much more high-speed railways in east China than the west. Moreover, inter-city subways connect many cities in the east but very few have been found in the west. Thus, actors in the East are expected to spend less time in travelling the same distance making it more convenient for frequent communications. In that case, travelling time may give better information about geographical diversification. Therefore, here we followed a prior study [90] and measured distance by travel time. Specifically, if the traffic time was less than 3 h, the geographic distance between cluster firm and its nonlocal subsidiaries was given a value of 1. If the traffic time was 3–12 h, the geographic distance equaled to 2. Similarity, the variable would get a value of 3 and 4 when travel time takes 12 h to one day and more than one day, respectively. By construction, the spatial distance between firms and its foreign subsidiaries would get the maximum value 5.

Social distance. Prior studies adopt the path length in a network [91] or number of links between two actors [47,92] to estimate social distance so as to capture the basic character of this indicator, i.e., the closeness of relationships. However, these measurements cannot guarantee validity since the role of each link has been assumed to be the same, which omits the fact that the success of innovation is also contingent on the duration of collaboration experience [93]. Distinct ties may generate dissimilar influence on innovation since they are different in frequency and duration of contacts. Firms with a long term interaction or co-work experience are more likely to build trust [70,71] and form shared

value or culture [75,76]. In advance, relationships between collaborators would become closer with the increase in duration of contacts. Following this logic, larger social distance implies fewer contacts and less experience of collaboration [94]. Thus, in this paper, social distance was measured by the inverse of average number of years that a firm had collaborated with its subsidiaries. The longer the duration of collaboration, the smaller the social distance.

### 3.2.3. Control Variables

To account for the effect of extraneous factors, we included a set of control variables considered in previous studies.

Technological proximity. Since collaborating with partners in similar technological fields can ease the innovation process [95,96], it is necessary to consider technological proximity in our model. Here we followed other researchers and used the same methodology to calculate technological proximity between two firms on the basis of the firms' technology fields [97,98].

Firm size. The size of firms is confirmed by studies as an influential factor for firms' innovative performance [83,99,100] and hence needs to be controlled. This variable is measured by the number of employees [101] and classified into five groups according to the file, i.e., notice on printing and distributing the standard provisions for the classification of small and medium-sized enterprises, issued by the Ministry of industry and information technology of the people's Republic of China in 2011, i.e., '<20', '20–50', '50–100 workers', '100–300' and '>300'. These groups were coded 1 to 5 respectively.

The development stage of firms. Researchers have found that firms in different stage of operation have distinct innovative performance [102]. Hence, it is necessary to design a variable to capture the maturity aspect of the firm. Based on widely accepted 5-stage classification, we, during our survey, asked the manager to consider the market share, age of firm, reputation and the scale of production, etc. so as to identify their firms' development stage clearly. Then, we also rechecked managers' answers by discussing with officers from High tech Zone Management Committee who were familiar with the history and situation of our sample firms. Based on the managers' response to our survey, we classified firms into five stages of development: 'seed stage', 'start-ups', 'growing stage', 'expansion stage' and 'mature stage'.

### 3.2.4. Methods

The following regression model was used to examine the impact of NMS and NRS on cluster firms' innovative performance:

$$INP = \alpha_0 + \beta_1 NMS + \beta_2 NRS + \beta_3 GED + \beta_4 SOD + \beta_5 SIZE + \beta_6 SDE + \beta_7 TPR + \varepsilon, \tag{1}$$

where INP stands for innovative performance; NMS represents nonlocal manufacturing subsidiaries and NRS is the number of nonlocal R&D subsidiaries. SOD is the social distance and GED denotes the geographical distance. SIZE, SDE and TPR represent firm size, stage of development and technological proximity, respectively.

To ascertain the moderating effect of geographical distance, we added interaction terms GED × NMS, SOD × NMS and GED × NRS, SOD × NRS to Model (1). Thus, Model (2) and Model (3) are expressed as follows:

$$INP = \alpha_0 + \beta_1 NMS + \beta_2 NRS + \beta_3 GED + \beta_4 SOD + \beta_5 SIZE + \beta_6 SDE + \beta_7 TPR + \beta_8 GED \times NMS + \beta_9 SOD \times NMS + \varepsilon. \tag{2}$$

$$INP = \alpha_0 + \beta_1 NMS + \beta_2 NRS + \beta_3 GED + \beta_4 SOD + \beta_5 SIZE + \beta_6 SDE + \beta_7 TPR + \beta_8 GED \times NRS + \beta_9 SOD \times NRS + \varepsilon. \tag{3}$$

In sum, we used Model (1) to examine the relationship between the focus independent variables and the dependent variable, and Model (2) and Model (3) to ascertain the moderating effect of geographical distance and social distance. Hierarchical regression analyses were employed to test our hypotheses.

## 4. Results

### 4.1. Descriptive Statistics

Table 3 reports the mean values, standard deviations and Pearson correlations between variables of interest.

**Table 3.** Descriptive statistics of variables and correlation matrix.

| Variables | Mean | Std. dev | INP | NMS | NRS | GED | SOD | SIZE | SDE | TPR |
|---|---|---|---|---|---|---|---|---|---|---|
| INP | 3.294 | 0.61 | 1 | | | | | | | |
| NMS | 0.684 | 0.468 | 0.735 *** | 1 | | | | | | |
| NRS | 2.52 | 1.024 | 0.683 *** | 0.583 *** | 1 | | | | | |
| GED | 1.57 | 0.608 | −0.643 *** | −0.62 *** | −0.411 *** | 1 | | | | |
| SOD | 2.37 | 0.850 | −0.533 *** | −0.413 *** | −0.51 *** | 0.371 *** | 1 | | | |
| SIZE | 1.65 | 0.801 | 0.121 | 0.039 | 0.122 | −0.067 | 0.005 | 1 | | |
| SDE | 2.89 | 0.554 | 0.115 | 0.057 | 0.026 | 0.043 | 0.09 | 0.11 | 1 | |
| TPR | 0.537 | 0.190 | −0.005 | 0.23** | −0.072 | 0.001 | −0.072 | −0.032 | 0.02 | 1 |

***$p < 0.01$, **$p < 0.05$, *$p < 0.1$

As can be seen from Table 3, the mean value of NMS shows that it was prevalent among cluster firms in our sample to have nonlocal manufacturing subsidiaries. In addition, the mean number of NRS a cluster firm held being between 2 and 3 confirms that it was a common phenomenon for cluster firms to distribute innovation sectors over sparse geographical space. The mean value and standard deviation of GED also reveal that the firms in our sample needed to spend 12 h on average to reach their nonlocal subsidiaries.

The correlation matrix in Table 3 indicates that there was a positive correlation between NMS and IPN. In other words, it seems that cluster firms could achieve greater innovative performance if they had developed nonlocal manufacturing segments. The number of NRS was also positively correlated with IPN. However, GED and SOD were negatively related to IPN. Thus, it seemed to imply that firms with closer, spatially or socially, nonlocal subsidiaries tended to have better innovative performance.

### 4.2. Regression Results

The regression results are shown in Table 4.

**Table 4.** Regression results.

| Variables | INP | | | | | | |
|---|---|---|---|---|---|---|---|
| Intercepts | −0.082 (−0.406) | 0.619 *** (4.834) | 0.012 (0.064) | 0.003 (0.018) | −0.107 (−0.398) | 0.021 (0.106) | 0.009 (0.046) |
| NMS | 0.506 *** (5.505) | | 0.379 *** (4.107) | 0.409 *** (4.229) | 0.387 *** (4.197) | 0.387 *** (3.790) | 0.383 *** (4.152) |
| NRS | | 0.426 *** (5.084) | 0.297 *** (3.612) | 0.356 *** (4.509) | 0.373 *** (4.487) | 0.300 *** (3.570) | 0.353 *** (3.481) |
| GED | −0.234 *** (−2.672) | −0.404 *** (−5.292) | −2.230 *** (−2.826) | −0.187 (−1.553) | −0.207 ** (−2.475) | −0.229 *** (−2.800) | −0.216 ** (−2.616) |
| SOD | −0.257 *** (−3.456) | −0.183 ** (−2.247) | −0.160 ** (−2.164) | | | −0.182 (−1.379) | −0.178 ** (−2.334) |
| NMS × GED | | | | −0.079 * (−1.750) | | | |

**Table 4.** *Cont.*

| Variables | INP | | | | | | |
|---|---|---|---|---|---|---|---|
| NRS × GED | | | | | 0.141 **<br>(2.078) | | |
| NMS × SOD | | | | | | −0.031<br>(−0.204) | |
| NRS × SOD | | | | | | | 0.093 *<br>(0.978) |
| SIZE | 0.070<br>(1.043) | 0.026<br>(0.373) | 0.039<br>(0.626) | 0.026<br>(0.399) | 0.052<br>(0.814) | 0.040<br>(0.633) | 0.045<br>(0.718) |
| SDE | 0.114 *<br>(1.686) | 0.136 *<br>(1.978) | 0.108 *<br>(1.727) | 0.109<br>(1.617) | 0.057<br>(0.873) | 0.107 *<br>(1.677) | 0.108 *<br>(1.725) |
| TPR | −0.140 **<br>(−2.010) | −0.075<br>(−1.091) | −0.143 **<br>(−2.217) | −0.135 **<br>(−2.012) | −0.139 **<br>(−2.155) | −0.143 **<br>(−2.205) | −0.135 **<br>(−2.082) |
| $R^2$ | 0.682 | 0.668 | 0.731 | 0.716 | 0.730 | 0.732 | 0.735 |
| Adjust $R^2$ | 0.656 | 0.640 | 0.705 | 0.688 | 0.704 | 0.701 | 0.705 |
| F | 25.479 *** | 24.107 *** | 27.628 *** | 25.572 *** | 27.446 *** | 23.853 *** | 24.280 *** |
| Obs. | 79 | 79 | 79 | 79 | 79 | 79 | 79 |

*** $p < 0.01$, ** $p < 0.05$, * $p < 0.1$, t value in parentheses, two tail tests.

Regression model of column 2 only included the NMS and control variable. Results in column 2 show that NMS had a significant effect on INP when the influence of NRS had not been considered. In column 3, results indicate the significant impacts of NRS on INP. Results of the full model, i.e., model (1), are summarized in column 4. A white test on this model shows that there was no obviously heteroscedasticity (Prob. Chi square = 0.0635 < 0.05). We also examined variance inflation factors (VIF) of model (1). The maximum value of VIF was 2.246 and the mean VIF value was 1.484, which was also below 10. In column 4, coefficients of NMS and NRS were both positive and significant ($p < 0.01$). In sum, results in Columns 2–4 indicate that the regression coefficient of NMS and NRS were positive and significant. Therefore, Hypotheses 1 and 2 were supported by our regression tests, i.e., cluster firms with nonlocal manufacturing subsidiaries and nonlocal R&D segments could achieve higher innovative performance. It is worth noting that GED and SOD were negatively related to INP. This finding was consistent with our conjecture that, geographically or socially, the closer the nonlocal subsidiaries were located, the greater innovative performance a cluster firm would achieve. While the regression results in the second column seemed to support a general negative impact of GED and SOD on INP, we still did not know if the effects of geographical distance and social distance were specifically related to which kind of nonlocal subsidiaries. In other words, it would be desirable to ascertain if cluster firms' innovative performance could be improved by having closer NMS or NRS or both. To detect the role of GED more precisely, we estimated Model (2) and Model (3) with interaction terms of NMS, NRS and GED. In the process of constructing interaction terms, we mean centered independent variables. The moderate effect of GED and SOD are summarized in Columns 5−8.

As shown in Columns 5 and 6, the coefficients of NMS × GED and NRS × GED were significant, which revealed that geographical distance negatively affected the relationship between NMS/NRS and cluster's innovative performance. Thus, Hypothesis 3a and Hypothesis 3b were supported. Columns 7 and 8 show that the coefficient of NMS × SOD was insignificant and NRS × SOD was positive and significant, indicating that nonlocal R&D subsidiaries would improve cluster firms' innovative performance if they were socially dissimilar with their holding companies. Hence, Hypothesis 4a and Hypothesis 4b were not supported by our regression results.

## 5. Discussions

*5.1. Theoretical Implications*

This paper aimed to detect and clarify the influence of two types of subsidiaries, NMS and NRS, on cluster firms' innovation and the moderating effect of distance. Based on a 79-firms sample from China, we confirmed the positive role of both NMS and NRS on their holding firms' innovation. Geographical distance negatively moderated the role of NMS and NRS while social distance only positively affected the role of NRS. These results provided novel insight into the effect of nonlocal subsidiaries on cluster firms' innovation and contributed to the literature in two ways.

First, this paper was the first research, to our knowledge, that considered both NMS and NRS in the theoretical model and provided empirical evidence for their role on cluster firms' innovation. The existing literature on cluster innovation pays a lot attention to the nonlocal knowledge linkage issues [51] but seldom clarifies roles of distinct nonlocal subsidiaries on their holding firms' innovative performance. Our research gives a better understanding about the contribution of two types of nonlocal subsidiaries to cluster firms' innovation. In addition to confirming the positive influence of NRS and providing empirical support to existing literature on knowledge diffusion, our findings stress that NMS are also important platforms for integrating valuable external experience and knowledge. Even if the main purpose of moving production segments is to exploit low cost advantage or to reach a broader market, cluster firms also become more innovative through the process of holding NMS. A possible explanation for this linkage may be that practical or tacit knowledge in workplaces could be absorbed and integrated by extra-cluster affiliations through co-working or problem solving, and as a result highly context-dependent knowledge and comparative systematic technology could be learned and transferred back to cluster firms. For example, Foxconn Technology Group, a Taiwanese international corporation, has set up a bunch of branch factories in mainland China since the 1980s and from which hundreds of patents have already been developed with Chinese partners [103]. Thus, NMS constitutes fertile ground for knowledge creation, which has long been under-recognized in traditional literature on extra-cluster innovation. Our study is a pioneering attempt to explore the effect of NMS on innovation during cluster firms' relocation process, an area largely neglected in prior studies. We hope our findings could stimulate more research on the linkage of cluster firms' R&D and relocated manufacturing activities.

Second, this paper also contributed to further understanding regarding the role of geographical distance and social distance on nonlocal learning and cooperation as well as topics about extra-cluster knowledge searching. Our findings suggest that deploying NMS/NRS in closer districts and sustaining loose contacts with NRS seem to be a better strategy for cluster firms' innovation. Unlike prior studies stressing the convenience of advanced information and communication technologies currently [104], we found that spatial distance still posed a significant negative effect on the role of NMS and NRS to their parent companies' innovation. It seems that knowledge both acquired from "learning by watching" and co-invention will be trapped in a specific area, which provides new evidence that the effect of geographical distance cannot be offset by temporary spatial proximity [39]. Our results also reveal that NRS will become more beneficial to cluster firms' innovation if they have somewhat less social contacts with their holding firms. Contrary to finding of prior studies in literature on social proximity [77], it seems that loosely connected relationships may grant NRS more autonomy in co-invention and that in turn enhances the prospect for their holding firms to successfully renew or complement the existing knowledge pool. Our research advances the literature on knowledge diffusion by clarifying the distinct effect of distance on the role NMS and NRS. Though the variety and the complicated nature of knowledge may hamper cluster firms to absorb and apply new technology from nonlocal subsidiaries, the benefit cluster firms can acquire could also be offset, partly or even completely in some cases, due to a lack of distinct sorts of proximities. We thought it was important to recognize the different effect of distance between NMS and NRS in cluster firms' choice for nonlocal development.

In sum, this study added to the literature on firms' relocation decision and nonlocal innovation, and expands present research by distinguishing geographical and social influence on the innovation contribution role of NRS and NMS. This study paved the way for further studies to unravel the reason behind differential distance influences.

## 5.2. Managerial Implications

Findings of this paper were also of relevance to managers and policymakers. Our results advise and inform cluster firms the proper strategies to improve their nonlocal portfolios. Specifically, management should realize that NMS not only could expand or renew cluster firms' production capability but also contributes to innovative performance. Regions that managers decide to relocate their extra-cluster manufacturing affiliations would be more desirable to be also places where local agents such as clients, customers as well as competitors can enrich subsidiaries' existing knowledge bases and are enthusiastic in technology sharing especially in tacit knowledge exchange in workshops. In this respect, cross cluster collaboration should be an important consideration for relocation-related decision making if the locus of the host region stands close to the target markets that firms plan to permeate. Policy makers in the home region also need to realize that it is not damaging that the manufacturing capacity is moving out as part of the strategic relocation process of local clusters. On the contrary, new channels for innovation could be developed, which may benefit the reconfiguration and upgrading of manufacturing capacity of local industries. Besides that, a cluster firm, which strives to relocate its R&D capacity, should care about its spatial closeness and social diversity with nonlocal subsidiaries. Our results reveal that closely connected NRS can improve holding firms' innovation performance. Moreover, managers of cluster firms should also take advantage of social dissimilarity (or diversity) of their NRS by granting decision-making autonomy on innovation-related issues.

In addition, our findings may provide insights to clusters, which are facing rigorous challenges during the upgrading process, such as those manufacturing clusters in the Pearl River Delta of China [17]. Those clusters are eager to reshape their industrial structure and phase out low-profit manufacturing sectors so as to develop and transform into an innovation-oriented district. In this respect, we suggest that policymakers could formulate better relocation policy to enlarge local innovative capacity from both the NMS and NRS. Cluster innovation policy should promote and provide incentives for firms to relocate their manufacturing subsidiaries to places not only with low cost advantage but also with ample collaborative opportunities and relationships with agents from demand side and supply side. If firms only focus on the cost advantages, they run the risk of losing innovative opportunities to be offered by NMS and NRS. Altogether, our findings imply that relocation strategies should consider the role of different types of subsidiaries on innovation and should not ignore the influence of geographical and social distances on distant knowledge transfer.

## 5.3. Limitations and Future Research Directions

There were some notable limitations of this study. First, this study might not be fully generalizable since our sample was from a single industry and the research setting was limited to cluster firms in China. Though our sample firms were scattered across two clusters and were reasonably representative, it might not be completely convincing to claim that our results were perfectly transferable to other industries. The general applicability of our results awaits to be tested under other industry settings.

Second, our results revealed that different types of subsidiaries had distinct influences on cluster firms' innovation and also exhibited diverse tolerance to the effect of distance. Therefore, it was reasonable to expect that firms with NMS or/and NRS might acquire a distinct benefit to their innovation. Nevertheless, the exact nature of the inter-relationships between these two types of subsidiaries needs to be studied further. For example, are they complements or substitutes with each other? Studies that compare the effect of these two different nonlocal portfolios in varying industrial contexts will be useful. The difference of learning and knowledge diffusion pattern between NMS and NRS also need

to be identified in future research so as to see through the co-innovated process of these two types of nonlocal subsidiaries with their holding firms.

Third, while our findings provided evidence about the existence of the relationships between nonlocal subsidiaries and innovative performance, it remains unclear about the dynamics of that relationship. That is, our study could not answer the following question: how do nonlocal subsidiaries learn from external organizations and how to integrate new knowledge with cluster firms' present knowledge base? Subsequent work should examine the dynamics and linkages between nonlocal learning, knowledge integration and innovation.

## 6. Conclusions

Based on a 79-firms sample from China, this study provided evidence that NMS and NRS had significantly contributed to innovative performance of their holding firms. The results here illustrated that knowledge learned by nonlocal subsidiaries would in turn induce further improvement in cluster firms' innovative performance. After adding interactive terms to capture the moderating effect of geographical distance and social distance, we found that the effect of NMS on cluster firms' innovative performance decreased by spatial distance but insensitive to social distance. Thus, how social distance would affect innovative capacity might not be a major concern when firms decide which location to put up their new plants. Contrary to the finding that distance moderated the influence of NMS on innovative performance, the effect of NRS increased by social distance but decreased with geographical distance. As a result, barriers to knowledge transfer induced by geographical distance had a significant effect on site selection decision of both NMS and NRS. Besides that, how to manage the social relationships with NRS would be crucial for managers' policy making.

**Author Contributions:** Conceptualization, X.X. and W.-C.H.; methodology, X.X.; validation, X.X.; formal analysis, X.X. and W.-C.H.; data curation, X.X.; writing—original draft preparation, X.X. and W.-C.H.; writing—review and editing, X.X. and W.-C.H.; funding acquisition, X.X.

**Funding:** This research was funded by the "National Natural Science Fund of China, grant number 71972154, NO. 71402137 and NO. 71672143", "The Humanity and Social Science on Youth Foundation of Ministry of Education, grant number 19YJA630091", "The Project Supported by Natural Science Basic Research Plan in Shaanxi Province of China, grant number 2019JM−235" and "The Youth Innovative Group Project of Xi'an University of Technology, grant number 105−451215006".

**Acknowledgments:** The authors are grateful to anonymous reviewers and editors for their comments and suggestions on this article.

**Conflicts of Interest:** The authors declare no conflicts of interest.

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
