# Peer review of "Does Distance Affect the Role of Nonlocal Subsidiaries on Cluster Firms’ Innovation? An Empirical Investigation on Chinese Biotechnology Cluster Firms"

_sustainability, doi:10.3390/su11236725_

Round 1

Reviewer 1 Report

Does Distance Affect the Role of Nonlocal Subsidiaries on Cluster Firms’ Innovation? An Empirical Investigation on Chinese Cluster Firms

General overview and structure

My impression is the topic, structure and approach is correct. The article has novelty and adds value to previous literature. However, in my opinion, the current version of the paper still has methodological limitations and should be improved in order to be published.

In my opinion, the title of the paper should consider the particular focus on biotech industry.

SECTIONS

Introduction

In order to identify the research gap, you mention (line 60 page 2) very few studies have investigated…..Please cite and justify that assertion. It happens the same for line 63 page 2.

The current introduction is well structured, but you should put more focus on the original contribution.

Literature review

Your theoretical reasoning for hypotheses in based on different theories, i.e.: knowledge diffusion theory, theory of agglomeration economy, social capital perspective, social proximity…Please, a further effort should be done in order to vertebrate the whole paper based on the most important theory or theories.

Literature review should be updated in order to include relevant new articles. References are not updated since 2016.

Methodology

Sample

Any further information regarding the source of your database will be welcomed: source, provider, population. Is it a random sample? You mention in the limitations, the sample is not representative from the whole population.

Have you controlled selection bias for sample? You should also put any further information regarding the treatment of missing values, as well as any test to control for non-respondents.

Additionally, in order to classify firms in small and medium companies, you should include in the text any definition to classify according to your criteria.

Regarding the dependent variable (innovation performance), how do you justify the composite index, is there any previous experience in literature? This is the most important measure used in the paper and should be stated in accordance to previous literature. You should consider any alternative way to measure innovation performance through a robustness test.

Independent variables

Please provide further justification for variables NRS and NMS according to previous literature.

Regarding the validity and goodness of fit in the model, you should include the statistical signification of the goodness of fit for the global model (F statistic).

Have your controlled also heterokedasticity and endogeneity in the model? So, White’s test or Breusch-Pagan test should be reported.

Please, provide more information to explain regression models of table 4, you present the results of table 4 with seven estimations.

The survey and questions should be included in the article in the appendix.

References

For example, review if new references should be included, for example:

Geldes, C., Heredia, J., Felzensztein, C., & Mora, M. (2017). Proximity as determinant of business cooperation for technological and non-technological innovations: a study of an agribusiness cluster. Journal of Business & Industrial Marketing, 32(1), 167-178.

Baycan, T., Nijkamp, P., & Stough, R. (2017). Spatial spillovers revisited: Innovation, human capital and local dynamics. International Journal of Urban and Regional Research, 41(6), 962-975.

Grillitsch, M., Martin, R., & Srholec, M. (2017). Knowledge base combinations and innovation performance in Swedish regions. Economic Geography, 93(5), 458-479.

Muller, E., & Peres, R. (2019). The effect of social networks structure on innovation performance: A review and directions for research. International Journal of Research in Marketing, 36(1), 3-19.

Reviewer 2 Report

Dear Authors,

Thank you for this opportunity to rAview your manuscript that explores the role of nonlocal manufacturing subsidiaries (NMS) and nonlocal  R & D subsidiaries (NRS) in influencing their parent firms' innovation.  This research is interesting and has the potential to make significant contributions to the literature.  However, your work in its current form requires significant improvement to enhance its academic value and practical implications, as there remain a number of critical flaws, particularly regarding the methodological rigor.

    First and foremost, although the research gaps were clearly addressed in the introduction section, the research design, particulalry regarding the measures you adopted, was not in response to your research questions.  More specifically, in the methodology section, the variables you created and selected couldn't answer the questions raised previously—I am actually skeptical of the reliability of your measures. 

    For example, on page 7, you mentioned that you used “”travel distance” to measure "geographic distance" between the parent firms and their NMS and NRS in China.  However, you have to better justify why and how this is more suitable than physical distance and territory border that have been widely adopted by many scholars.   Moreover, using the "number" of all kinds of R & D subsidiaries may not be appropriate for measuring the degree of R& D collaboration with their parent firms, as different types of R & D subsidiaries may perform very different functions.    

    On page 8, you claimed to measure social distance by the inverse of average number of years that a firm has collaborated with its subsidiaries.  This is not convincing to me.  I would suggest you to better justify the theoretical rationale. 

    In addition, using the managers' self-report data to characterize firm innovation performance and the development stage of firms (i,e., your control variable) is lack of objectivity.  It is also not very adequate to classify firm sizes into five groups merely by the number of employees— You have to better justify why the scores (i.e., 1 to 5) were given.

    Considering the lack of appropriateness and reliability of the variables used in the regression analyses, I am a bit skeptical of your results.  In terms of the theory section, I feel that your current hypotheses development may be too descriptive—I would suggest you to further strengthen the theoretical underpinning.

    Overall, despite quite a few interesting ideas, I would suggest you to conduct a fundamental rethinking to re-design the whole research due to the serious methodological concerns illustrated above.  Hope my comments are helpful.

Reviewer 3 Report

Dear Authors,
I have a few concerns on this article:

(1) The aim of the paper should be defined more precisely in the Introduction section and be consistent with the presented results.
(2) The Therory and Hypotheses section seems to be limited and have a limited number of current research in the fields.
(2) Resultes in <Table 4> Columns 5-5 indicate that the t-valueof NRS×GED variable are 2.078 and significant. It indicates significance at the 5% level, but you mentioned not siginificant.
(3) The t-value of TPR variable are 1.553 and insignificant, but Table 4 showen **(5% level).
(4) It is recommended to show the coefficient and t-stat. of the intercept.
(5) The implications should be better presented and linked to the literature reviews and develop the limitations of your research.

Round 2

Reviewer 1 Report

Authors have done an important effort to improve the manuscript according to suggestions provided.

Reviewer 2 Report

Dear Authors,

I can see you have put great effort into modifying your research as per the reviewers' comments and this revision indeed shows significant improvement.   
Although I am still a bit skeptical of the measures you used, I feel this version is well-structured and well-written.  Hence, I recommend to accept your manuscript in its current form for publication.